# *Trichoderma* spp.-Related Pneumonia: A Case Report in Heart–Lung Transplantation Recipient and a Systematic Literature Review

**DOI:** 10.3390/jof9020195

**Published:** 2023-02-02

**Authors:** Carlo Burzio, Eleonora Balzani, Giorgia Montrucchio, Anna Chiara Trompeo, Silvia Corcione, Luca Brazzi

**Affiliations:** 1Department of Anesthesia, Intensive Care and Emergency, Città della Salute e della Scienza di Torino Hospital, 10126 Torino, Italy; 2Department of Surgical Science, University of Turin, 10124 Torino, Italy; 3Department of Medical Sciences, Infectious Diseases, University of Turin, 10124 Turin, Italy; 4School of Medicine, Tufts University, Boston, MA 02111, USA

**Keywords:** emerging fungal infections, *Trichoderma* spp., heart–lung transplantation, cardiac surgery, critical care

## Abstract

Opportunistic and hospital-acquired infections are common among recipients of solid organ transplantation. New pathogens are increasingly reported in the intensive care unit (ICU) population. We report a case of a patient who developed *Trichoderma* spp.-related pneumonia (TRP) after heart–lung transplantation. In the absence of antifungal susceptibility testing, TRP was confirmed by histological examination, and empirical therapy with voriconazole and caspofungin was swiftly initiated. Complete resolution of pneumonia was obtained after prolonged combination therapy. Given the lack of guidelines, we conducted a systematic review to elucidate the diagnostic and therapeutic strategies to apply during *Trichoderma* infection. After deduplication and selection of full texts, we found 42 articles eligible for the systematic review. Pneumonia seems to be the most common clinical manifestation (31.8%). The most used antifungal therapy was amphotericin B, while combination therapy was also reported (27.3%). All the patients were immunocompromised except for one case. Despite the rarity of *Trichoderma* spp. infection, the increase in invasive fungal infections is of growing importance in ICU, considering their impact on mortality and the emergence of antifungal resistance. In the absence of prospective and multicenter studies, a review can provide useful insight regarding the epidemiology, clinical manifestations, and management of these unexpected challenges.

## 1. Introduction

Opportunistic and hospital-acquired infections can occur in solid organ transplantation (SOT) recipients. Among them, patients with the highest rate of infectious complications are those undergoing lung transplantation (LTx), especially in the intensive care unit (ICU), where invasive fungal infections (IFIs) are becoming increasingly common [1,2,3,4].

IFIs are a major contributor to nosocomial infection in critically ill patients. Most of them are attributable to *Candida* spp. (principally bloodstream or abdominal infections) and *Aspergillus* spp., representing an important cause of morbidity and mortality and being associated with a longer duration of ICU stays [5]. The incidence of IFIs has risen over recent decades. It is continuing to grow both in immunocompetent and immunocompromised hosts, as recently highlighted during the COVID-19 pandemic [6,7]. At the same time, the broadened use of systemic antifungal drugs makes them quite a common component of antimicrobial therapy, with the development of resistance now deserving antifungal stewardship strategies [8,9]. Considering the important limitation in terms of diagnostic tools and turn-around time of traditional tests, risk factor evaluation remains pivotal to guide clinical orientation to treatment [10]. The most common risk factors characterizing ICU admission and predisposing LTx recipients to opportunistic infection are immunosuppressive and immunomodulatory therapy, loss or impairment of cough reflex, CVM co-infection, ischemia of the anastomosis, and ICU-acquired weakness [11,12]. While the best-known opportunistic fungal infections in LTx recipients are those related to *Aspergillus* spp., infections due to new pathogens, such as molds and filamentous fungi, are on the rise.

*Trichoderma* spp. is an omnipresent mold, usually present in the soil. Under normal conditions, it does not cause active infections in healthy human hosts [13]. However, nine *Trichoderma* species have been reported to cause human infection, especially in immunocompromised hosts [14].

Starting from the clinical case of a patient who underwent an en bloc heart–lung transplant and developed a *Trichoderma* spp.-related pneumonia (TRP) in the first postoperative period despite the antifungal prophylaxis, we conducted a systematic review of the literature to describe clinical characteristics, confirm diagnostic methods, and revise available treatment options and outcomes.

### The Clinical Case

A 42-year-old woman was admitted to the cardiosurgical ICU after a heart–lung transplant. Her past medical history was characterized by Eisenmenger Syndrome and patent ductus arterious diagnosed at the age of 3. Her clinical conditions remained stable until the age of 38 when a progressive and symptomatic decline in pulmonary function occurred, and pulmonary fibrosis was diagnosed.

Due to the worsening of her pulmonary and cardiac conditions (increase in pulmonary hypertension and right ventricular dilatation), she underwent a combined heart–lung transplant. Her intraoperative course was complicated by prolonged cardiopulmonary bypass and graft ischemic time (cardiopulmonary bypass time of 285 min and graft ischemic time of 190 min) and by difficulty weaning from the cardiopulmonary bypass itself.

Antimicrobial prophylaxis was started according to our local protocol, including inhaled amphotericin B (amphotericin B lipid complex, 20 mg once daily) and oral nystatin (1 MIE oral drops, TID). Our protocol also included antibacterial prophylaxis (ceftazidime, IV 1 g TID, for the first 72 h), *Pneumocystis jirovecii* prophylaxis (Trimetoprim-Sulfamethoxazole, PO, once daily), and *Citomegalovirus* (CMV) prophylaxis according to CMV status (Valganciclovir, PO, 900 mg once daily).

After the patient was successfully weaned from the inotropic support end and extubated on postoperative day 5, her clinical conditions worsened on the ninth postoperative day due to the onset of moderate hypoxemia and clinical signs of pump failure. Leukocytosis appeared associated with an increase in and worsening of tracheal secretion, while a chest X-ray confirmed a left lower lobe consolidation. Additionally, her renal function progressively declined.

Empirical antibiotic coverage with vancomycin and ceftazidime was then initiated. Simultaneously, a chest CT scan confirming parenchymal consolidation and an extensive ground-glass pattern involving both inferior lobes and the lingula and bronchoalveolar lavage (BAL) were obtained. BAL did not find any bacterial growth, but a positive cell culture for *Trichoderma* spp. and a BAL-Galactomannan assay of weak positivity (EIA 0.60 S/CO) were detected. Hence, antifungal combination therapy with voriconazole and caspofungin was started. A cytological examination was also found positive for hyphal growth. Repeated BALs were performed, as well as culture examination of mucous plug; microscopic examination confirmed *Trichoderma* spp. and hyphal growth in all the samples, but our laboratory was unable to grow viable cultures, identify *Trichoderma* species, or perform antimicrobial susceptibilities testing (AST).

After 34 days of antifungal therapy, the resolution of the pneumonia was observed, as confirmed by clinical and chest CT improvement and by a negative BAL culture. Parenteral antifugal therapy was therefore discontinued, and prophylaxis with inhaled amphotericin B alone was maintained. Following surveillance cultures, including subsequent BALs and BAL-Galactomannan assays, were negative for *Trichoderma* or fungal detection.

The patient suffered ICU-acquired weakness and difficulty weaning from mechanical ventilation. Her clinical course was further complicated by MDR *Klebsiella pneumoniae* colonization and surgical site infection. Relapsing septic shock despite surgical debridement and end-stage renal failure ensued. The patient died from *K. pneumoniae*-related septic shock 6 months after transplantation.

While neither *Trichoderma* nor respiratory failure were direct causes of death, TRP was an early complication that required a prolonged ICU stay, thus hindering the postoperative clinical course.

## 2. Materials and Methods

### 2.1. Research Approach

The systematic review protocol was drafted using the Preferred Reporting Items for Systematic Reviews and Meta-analysis Protocols (PRISMA-P), which was revised by the members of our cardiac surgery research team, and the Preferred Reporting Items for Systematic Reviews and Meta-Analyses (PRISMA) statement [15]. The final protocol was prospectively registered with the Open Science Framework on 5 February 2022 (https://osf.io/n63ps, accessed on 1 January 2023).

### 2.2. Eligibility Criteria and Information Sources

To be included in the review, papers had to focus on human *Trichoderma* infection. Peer-reviewed articles were included if they were randomized-controlled trials (RCTs) or non-RCT (NRCTs), including case reports, and case series published in English or Italian and retrieved using Pubmed, Scopus, and Embase databases. The search was implemented with the use of registries (clinicaltrials.gov, accessed on 1 January 2023) and gray literature searches (http://greylit.org/, accessed on 1 January 2023).

### 2.3. Search Strategy

Literature search was carried out using the following search strategies:Pubmed: (“trichoderma”[MeSH Terms] OR “trichoderma”[All Fields] OR “trichodermas”[All Fields]) AND ((“human s”[All Fields] OR “humans”[MeSH Terms] OR “humans”[All Fields] OR “human”[All Fields]) AND (“infect”[All Fields] OR “infectability”[All Fields] OR “infectable”[All Fields] OR “infectant”[All Fields] OR “infectants”[All Fields] OR “infected”[All Fields] OR “infecteds”[All Fields] OR “infectibility”[All Fields] OR “infectible”[All Fields] OR “infecting”[All Fields] OR “infection s”[All Fields] OR “infections”[MeSH Terms] OR “infections”[All Fields] OR “infection”[All Fields] OR “infective”[All Fields] OR “infectiveness”[All Fields] OR “infectives”[All Fields] OR “infectivities”[All Fields] OR “infects”[All Fields] OR “pathogenicity”[MeSH Subheading] OR “pathogenicity”[All Fields] OR “infectivity”[All Fields]));Embase, Scopus, clinicaltrials.gov, and grey literature: (‘trichoderma’/exp OR trichoderma) AND (‘human infection’ OR ((‘human’/exp OR human) AND (‘infection’/exp OR infection))).

### 2.4. Selection and Data Collection Process

Search results were exported to EndNote V.X9 (Clarivate Analytics, Philadelphia, PA, USA). Duplicates were automatically removed. The review process was conducted in two steps consisting of evaluating the titles, abstracts, and then full text of all publications identified by our searches for potentially relevant manuscripts. For both levels, two authors (C.B. and E.B.) screened the articles, with conflicts resolved by consensus and discussion with other reviewers (A.C.T. and L.B.). A planned Excel spreadsheet was used to extract data (patients’ characteristics, year of publication, country, type of pathogen, therapy, and outcome measures).

### 2.5. Data Collection Process and Data Synthesis

We considered as variables the authors, the year of publication, the country where the article was published, the species of fungus that caused the infection, the main pathology from which the patient suffered, the clinical manifestation of the patient’s fungal infection, the treatment of choice, and the outcome.

### 2.6. Risk of Bias Assessment

Since we found only case reports and case series, it was impossible to apply the risk of bias assessment using the Risk Of Bias In Non-randomized Studies of Interventions (ROBINS-i) tool. Since these were descriptive studies of an infection and therefore not of a specific intervention, we did not deem it appropriate to perform a risk of bias assessment. Where the result has not been reported, we have highlighted this in the results table. Similarly, we decided not to apply a summary of evidence using the Grading of Recommendations, Assessment, Development and Evaluations (GRADE) model for the same reason: (1) there is a lack of comparison group, and (2) the cases are not groupable due to the geographic and temporal uniqueness of each case.

## 3. Results

### 3.1. Search Results

The systematic literature search performed on 11 October 2022 retrieved 386 results. None of them were found from grey literature or clinical trials registers. After deduplication, 72 studies were considered eligible for the full-text evaluation. A total of 42 of them were selected for full-text review and included in the systematic review (Figure 1). The reason for excluding the other full texts is that they did not discuss *Trichoderma*-related diseases but rather other fungal infections.

A total of 50% of the articles were published between 1969 and 1999, 22.5% between 2000 and 2010, and 27.5% from 2011 to the present. The countries of provenance are Spain at 20.5%, France at 18.2%, USA at 15.9%, Japan at 9.1%, Italy at 6.8%, and Saudi Arabia at 2.3% (Table 1).

### 3.2. Trichoderma and the Related Disease

*Trichoderma* spp. infection is reported in 25% of cases, while *Trichoderma* was typed in the remaining 75%. A total of 45.5% of the infections were due to *T. longibrachiatum* and 13.6% to *T. viride*.

In 31.8% of cases, pneumonia was the highlighted clinical manifestation. We found 12 papers (14 cases) specifically referring to TRP. In all of these cases, TRP affected immunocompromised hosts and in most of them, namely 78.6%, turned out to be non-fatal. However, it should be noted that in 14.3% of cases, it was not possible to determine the outcome, as it was not reported by the authors in the abstract, and the full paper was written in Japanese [21,25].

Other clinical manifestations are less common (Figure 2). *Trichoderma* resulted in peritonitis in 22.7% of cases, and 40% of these subjects survived. Of these patients, 10/12 were patients undergoing outpatient peritoneal dialysis, one was a patient with end-stage renal failure, and one was a patient undergoing kidney transplantation. Intra-abdominal infections without peritonitis accounted for 6.8% of cases, all of them being post-abdominal surgery patients.

Skin infection by *Trichoderma* accounted for 9.1% of cases, all of them affecting immunocompromised hosts, with a survival rate of 75%.

The reported cases of endocarditis are relatively recent, ranging from 2000 to 2016, representing 6.8% of cases, with 100% mortality. Of these, one patient was receiving home parenteral nutrition [45], one was a patient undergoing aortic surgery [34], and one was a patient undergoing ICD implantation complicated by pneumothorax [48].

There are two cases of brain infection with brain abscess by *Trichoderma*, and they were reported in 1995 by *T. longibrachiatum* in Germany in a patient affected by leukemia [24] and in 1999 by *T. harzianum* in Spain in a kidney transplant patient [32]. Of these, the patient with brain abscess and leukemia, treated with amphotericin B, itraconazole, and surgical debridement, had survived. A case of *Trichoderma* spp.-related encephalitis was reported in a patient with AIDS, treated with amphotericin B, who survived [35]. Disseminated *Trichoderma* infection is reported in 6.8% of cases, with a survival rate of 33%. Two of three patients with disseminated *Trichoderma* infection were bone marrow transplant patients [23,31].

Two cases of sinusitis have been reported, both from *T. longibrachiatum,* in a liver transplant patient and in an immunocompetent patient [30,46]. Of these two cases, both patients survived with treatment with amphotericin B and surgical debridement. The liver transplant patient had received combination therapy with amphotericin B and itraconazole.

Overall, according to the literature, 27.3% did not survive *Trichoderma* infection even though, in 4.5% of the articles, the outcome is not reported.

### 3.3. Diagnostic Tests and Antifungal Susceptibility Testing

A summary of the diagnostic approach and antifungal susceptibility testing is highlighted in Table 2.

*Trichoderma* infection was often diagnosed by culture of body fluid involved, being peritoneal fluid in peritonitis or BAL in pulmonary infection. Bioptic examination was rarely employed outside skin infections, in which case it remained the preferred diagnostic test. Regarding AST, data reviewed confirm that *Trichoderma* spp. maintains elevated MIC toward commonly employed azoles, such as fluconazole and ketokonazole. Most samples analyzed instead reported a significantly low MIC (<1 μg/mL) regarding echinocandins, in most cases caspofungin (MIC < 1 μg/mL in 10 out of 13 samples, 76.9%). Variable resistance patterns can be shown for voriconazole (MIC < 1 μg/mL in nine out of sixteen samples, 56.5%) and amphotericin B (MIC < 1 μg/mL in 10 out of 24 samples, 41.6%).

Furthermore, the diagnostic approach was often more complex in TRP cases. Out of twelve cases, only seven cases were diagnosed by BAL culture and microscopic examination. In two cases with clear clinical respiratory involvement and sepsis, blood cultures were diagnostic for *Trichoderma* spp.; both of them were neutropenic patients. In one case, transbronchial biopsy was deemed necessary and isolated hyphal growth. Galactomannan assay was performed in only three cases, in one case from BAL material, in a second case from serum analysis, and the last from an unspecified sample; one Serum-Galactomannan assay was positive (EIA = 1.3), while the other two samples were reported negative. Species identification based on PCR amplification and partial DNA sequencing was attempted in six case reports. *Trichoderma longibrachiatum* was identified in five of these cases. In one case, sequence analysis confirmed *Trichoderma* species without subspecies identification.

Antifungal susceptibility testing was performed in eight cases of TRP, although the number of antifungal drugs tested varies between studies. Antifungal susceptibility was determined with European Committee on Antimicrobial Susceptibility Testing (EUCAST) standardized methodology. As expected, *Trichoderma* was vastly resistant to fluconazole and itraconazole, with MIC often >32 μg/mL. Voriconazole showed MIC < 1 μg/mL in four cases and MIC ≥ 1 μg/mL in three cases. Amphotericin B exhibited MIC < 1 μg/mL in three cases and MIC ≥ 1 μg/mL in four cases. Five cases tested Caspofungin susceptibility, and *Trichoderma* showed clear in vitro susceptibility in all of them, with MIC < 0.5 μg/mL. Finally, one case reported in vitro susceptibility to amphotericin B and resistance to both voriconazole and caspofungin, without disclosing any MIC value.

### 3.4. Antifungal Management

Antifungal therapy varies among observed cases, depending on timing and site of infection. Most cases relied on amphotericin B, either alone (fifteen cases, 34.1%) or in combination with an azole (eight cases, 18.8%). In eight cases (18.2%), azoles (either itraconazole, ketoconazole, fluconazole, miconazole, or voriconazole) alone were the treatment of choice, while in three cases (6.8%), echinocandin alone was administered. In four cases (9.1%), a combination therapy with echinocandin and voriconazole was deemed necessary. Lastly, in a recent clinical case involving fungal pericarditis, isavuconazole was administered. One paper does not specify the type of antifungal drug used [57].

Concomitant source control procedures, including surgical debridement and catheter removal, were used in 14 cases (35.9%), none of them involving pulmonary infection.

Regarding TRP, the treatment of choice was either amphotericin B plus azole (two patients, 14.3%), amphotericin B in monotherapy (four patients, 28.6%), or echinocandin plus voriconazole (two patients, 14.3%). Amphotericin B plus voriconazole plus caspofungin combination therapy was used in only two cases (14.3%).

Patients who developed *Trichoderma* spp. infection were reported to be immunocompromised (sixteen cases), onco-hematological (fifteen cases), transplanted (seven cases, two of them lung transplantation), and AIDS (two cases). Among transplant patients, only one lung developed *Trichoderma* pericarditis [34]. Only in one case [46] did *Trichoderma* infection affect an immunocompetent patient; in this case, the clinical manifestation was a sinusitis which ended with a resolution of the infection by surgical debridement and antifungal topical therapy.

## 4. Discussion

Evidence from the scientific literature seems to suggest that *Trichoderma* spp. represents a re-emerging fungal infection (Table 1). The prolonged use of antifungals and antibacterials in agriculture and medicine has indeed altered the global microbiome, leading to the growth of drug-resistant fungi [58] such as *Candida auris*, *Histoplasma*, *Cryptococcus*, and *Aspergillus* spp. [2].

Specific risk factors for *Trichoderma* infection are difficult to detect due to the paucity of cases. The literature review confirmed that *Trichoderma* spp. infections occur mostly in immunocompromised patients (Table 1). Our patient showed several risk factors for invasive aspergillosis, as her medication regimen included a steroid for immunosuppressive therapy, Anti-Thymocyte Immunoglobulin as induction therapy; notably, no specific environmental factor could be found in the medical history of either the organ donor or the patient other than prolonged pre-transplantation hospitalization of the recipient.

Patients undergoing SOT and hematopoietic stem cell transplantation are likely to develop emerging IFI due to immunosuppressive and immunomodulatory therapy [12]. Furthermore, the widespread use of anti-*Aspergillus* prophylaxis in SOT recipients may exert selective pressure toward uncommon pathogens [59], such as pathogens generally found in soil (e.g., phaeohyphomycosis), *Fusariosis*, *Scedosporium*, and *L. prolificans* [60]. Given that *Trichoderma* infection has been found in SOT recipients, namely renal and liver recipients (five cases), it is not surprising to find it in heart–lung recipients as well. Pneumonia is the most reported clinical manifestation, considering both the tendency of these patients to experience pulmonary complications and the fact that the lung is the most common site of infection for uncommon pathogens, such as *Trichoderma* spp.

There is no doubt that the diagnostic evaluation of TRP is particularly challenging. While typical fungal colonization or disease such as pulmonary Aspergillosis are defined by solid evidence-driven diagnostic criteria [61], the diagnosis of non-*Aspergillus* invasive mold infection (NAIMI) is based on extrapolated data or expert opinions. Even if the distinction between colonization and active infection is crucial in opportunistic fungal infections [62], this is lacking for NAIMIs [63]. The literature suggests the early use of CT scan, both to diagnose and delineate the disease and to guide invasive procedures aimed at obtaining deep tissue samples (including BAL). Specific markers to identify NAIMI infections [59] are often unavailable [12], and most laboratories do not have adequate expertise to detect and identify rare pathogens. BAL galactomannan (BAL-GM) can be used to evaluate and diagnose Aspergillosis [61]. However, its application in other filamentous fungi infections is controversial, since cross-reactivity of the BAL-GM assay with both *Trichoderma pseudokoningii* and *T. longibrachiatum* infections has been reported, with a moderate increase in its values [62]. The weak positivity of the BAL-GM assay that we found is in line with evidence found in the literature. The use of nucleic acid amplification tests (polymerase chain reaction and sequencing on fungal RNA genes) has been described as highly specific and sensitive [12], but their availability is still limited, as in the case of our institution, where is not available. A specific histological investigation to evaluate the presence of hyphae in the BAL confirmed a pulmonary infection, as recommended for unspecified NAIMI.

The clinical status of critically ill patients is frequently not conducive to early lung tissue biopsy. In the case described above, it was not deemed appropriate due to the high risk of reported complications in mechanically ventilated patients. Conversely, the use of BAL together with histological investigation might offer an adequate risk/benefit ratio, being repeatable and easy to perform. From a therapeutic point of view, uncommon IFIs might be affected by antifungal resistance [2]. Antifungal susceptibility testing (AST) has not been routinely performed in all cases of pulmonary *Trichoderma* infections, mainly due to the difficulty of isolating and growing filamentous fungi (as in the case described) [49,63]. Furthermore, in NAIMI, AST is not recommended, as there are no breakpoints and no established correlation between minimal inhibitory concentration (MIC) and clinical outcome for most uncommon fungal infections [64,65]. This recommendation has been proposed since a correlation between MIC and clinical outcomes in opportunistic mold infections is missing, while a discrepancy between in vitro AST and in vivo infections has been previously reported and suggests poor clinical relevance [59].

Nevertheless, AST for *Trichoderma* infections is being widely utilized in clinical practice (Table 2), often assuming clinical breakpoints for unspecified mold infection. Our literature review suggests that *Trichoderma* maintains high in-vitro resistance for azoles, showing MIC often >32 μg/mL. High (>1 μg/mL) MIC values are often reported for voriconazole and amphotericin B as well. Echinocandins instead exhibited MIC < 1 μg/mL in most of the reported cases. These results agree with a previous study regarding in vitro susceptibilities of isolated *Trichoderma*, including both human and animal samples (however, 40% of samples were collected from the human respiratory tract), which reported mostly low MIC for voriconazole and echinocandins [14].

In the absence of a clinical breakpoint, AST-adjusted antifungal therapy is recommended with low-quality evidence [66]. Although in some cases AST has been obtained and used to guide antifungal therapy [36,39,40,56], no recommendation regarding AST in *Trichoderma* spp. can be found in the literature.

Treatment for *Trichoderma* spp. usually consists of amphotericin B or azole (Table 1). In our case, the patient was already receiving amphotericin B aerosol for prophylaxis, raising concerns about a possible breakthrough infection. Possible adverse effects caused by an amphotericin B intravenous course were also considered. Combination therapy with voriconazole and an echinocandin was described to treat *Trichoderma*-related pulmonary infections with a favorable therapeutic index [40], as in vitro sensitivity of most *Trichoderma* spp. to both drugs has been reported in previous studies [14] as well as in our literature review (Table 2). In our patient, caspofungin–voriconazole combination therapy was chosen, considering the possibility of amphotericin B resistance in the context of breakthrough infection and the absence of AST, as well as the patient’s comorbidities. Due to concomitant calcineurin inhibitor therapy, a voriconazole-monitoring-driven approach was conducted, as recommended by guidelines [67,68].

Another possible option reported in the literature is represented by liposomal amphotericin B, previously described to treat TRP without further impairment of renal function [49], as well as to treat other *Trichoderma* infections (Table 1). In animal models, disseminated *Trichoderma* infection has shown peculiar resistance to most antifungal agents, including amphotericin B and voriconazole; however, combination therapy was not tested in this study [69]. According to the literature, both combination treatment and amphotericin B monotherapy were previously found effective to eradicate the infection.

From the data extracted in the literature, we can suggest with this review to carefully evaluate according to the risk factors (e.g., outpatient peritoneal dialysis patient, immunocompromised patient with unspecified/unresolving pneumonia) the possibility of re-emergent fungal infection, such as *Trichoderma*. In the absence of universally recognized definitions and diagnostic criteria, careful prompt examination and investigation, based on both a CT scan and BAL with nucleic acid amplification tests, if available, are crucial for early diagnosis. Although laboratory MIC cut-offs are still lacking, AST might be encouraged due to high MIC reported for many antifungal medications, especially considering the global increase in fungal resistance and the need for antifungal stewardship strategies. In the absence of AST, local susceptibilities as well as patient characteristics, including previous prophylaxis or therapies, should be carefully considered to tailor the appropriate antifungal therapy. Both amphotericin B monotherapy and antifungal combination with echinocandin plus voriconazole is reported as effective in *Trichoderma* spp. infections and can be viable therapies according to the clinical context.

## 5. Conclusions

Although rare, IFIs caused by uncommon fungi, such as *Trichoderma* spp. and other NAIMI, are increasingly reported in critically ill patients, especially in SOT recipients, presenting challenges in terms of clinical suspicion, diagnostic methods, and treatment options.

Large and definitive clinical trials involving *Trichoderma* human infection are unlikely due to the paucity of cases. The topic of re-emerging fungal infections also makes the possibility of an individual patient meta-analysis difficult because of the great differences in the geographic and historical context.

In case of future reports in the literature, we suggest improving the available evidence by reporting in more detail the MIC for antifungals, days of therapy, the outcome, and the concomitant complications developed by the patient.

## Figures and Tables

**Figure 1 jof-09-00195-f001:**
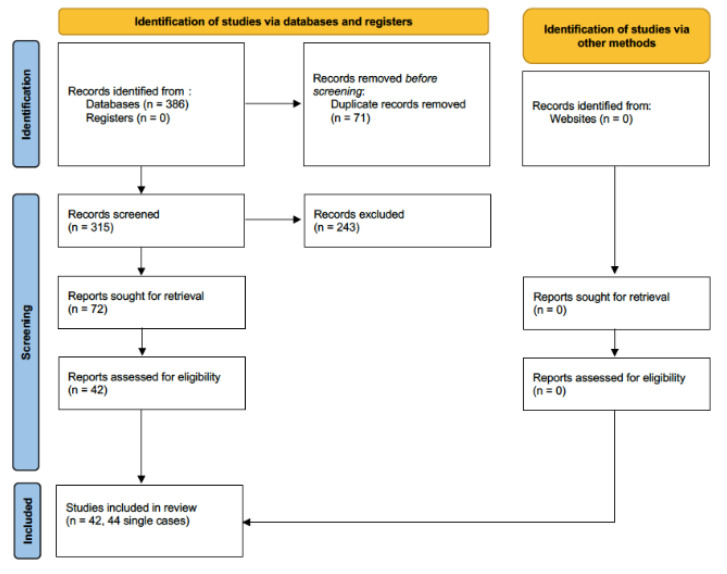
PRISMA flow diagram.

**Figure 2 jof-09-00195-f002:**
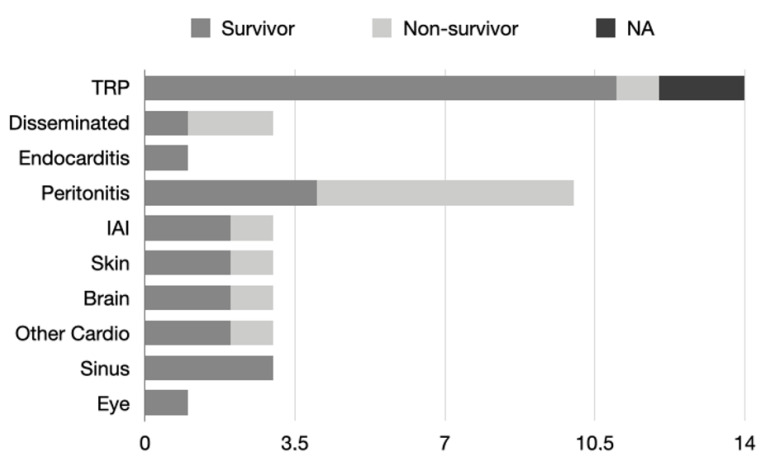
The graph shows the number of cases in the literature divided by type of *Trichoderma* infection developed by patients. Each bar is divided according to patient outcome.

**Table 1 jof-09-00195-t001:** Summary of findings.

Author	Year	Country	Fungal Species	Main Pathology	Clinical Manifestation	Treatment of Choice	Outcome
Robertson et al. [16]	1969	UK	*Trichoderma* spp.		Disseminated	AmB	Infection resolution
Escudero Gil et al. [17]	1976	Spain	*T. viride*	Acute myeloid leukemia	Pulmonary infection	Surgery	Infection resolution
Escudero Gil et al. [17]	1976	Spain	*T. viride*	Acute myeloid leukemia	Pulmonary infection	AmB + VOR + CASP	Infection resolution
Loeppky et al. [18]	1983	USA	*T. viride*	Renal failure secondary to amyloidosis	Peritonitis	Peritoneal catheter removal + AmB	Infection resolution
Ragnaud et al. [19]	1984	France	*T. koningii*	CAPD	Peritonitis	Peritoneal catheter removal + miconazole	Infection resolution
Jacobs et al. [20]	1992	Belgium	*T. viride*	Liver transplantation	Perihepatic hematoma and peritoneal fluid	AmB + FLU	Death
Imokawa et al. [21]	1993	Japan	*Trichoderma* spp.	Not stated	Pulmonary infection	NK	NK
Tanis et al. [22]	1995	The Netherlands	*T. longibrachiatum*	Renal transplantation	Peritonitis	AmB	Death
Gautheret et al. [23]	1995	France	*T. pseudokoningii*	Bone marow transplantation	Disseminated	AmB + 5-FC	Death
Seguin et al. [24]	1995	France	*T. longibrachiatum*	Leukemia	Brain abscess	AmB, ITR + surgery	Infection resolution
Kawaguchi et al. [25]	1995	Japan	*T. viride*		Pulmonary infection	NK	NK
Guiserix et al. [26]	1996	France	*T. harzianum*	CAPD	Peritonitis	Peritoneal catheter removal + KET	Death
Campos-Herrero et al. [27]	1996	Spain	*T. koningii*	CAPD	Peritonitis	KET	Death
Munoz et al. [28]	1997	USA	*T. longibrachiatum*	Aplastic anemia	Skin infection	AmB	Infection resolution
Bren et al. [29]	1998	Slovenia	*Trichoderma* spp.	End-stage renal failure and CAPD	Peritonitis	KET	Infection resolution
Furukawa et al. [30]	1998	USA	*T. longibrachiatum*	Small-bowel and liver transplantation	Invasive sinusitis	AmB + ITR, surgery	Infection resolution
Richter et al. [31]	1999	USA	*T. longibrachiatum*	Allo-HSCT	Disseminated	AmB, ITR	Death
Guarro et al. [32]	1999	Spain	*T. harzianum*	Renal transplantation	Brain abscess	None	Death
Rota et al. [33]	2000	Italy	*T. pseudokoningii*	CAPD	Peritonitis	Peritoneal catheter removal + AmB	Infection resolution
Bustamante-Labarta et al. [34]	2000	Argentina	*Trichoderma* spp.	Aorta surgery	Endocarditis	Antifungal, graft replaced	Infection resolution
Amato et al. [35]	2002	Portugal	*Trichoderma* spp.	AIDS	Encephalitis	AmB	Infection resolution
Chouaki et al. [36]	2002	France	*T. longibrachiatum*	Liver transplantation	Perihepatic hematoma	Surgery	Infection resolution
Chouaki et al. [36]	2002	France	*T. longibrachiatum*	Lung transplantation	Pulmonary infection	Amb	Death
Myoken et al. [37]	2002	Japan	*T. longibrachiatum*	Lymphoma	Necrotizing stomatitis	AmB, ITR	Death
Esel et al. [38]	2003	Turkey	*Trichoderma* spp.	CAPD	peritonitis	AmB	Death
De Miguel et al. [39]	2005	Spain	*T. viride*	Acute myeloid leukemia	Pulmonary infection	AmB and VOR	Infection resolution
Alanio et al. [40]	2008	France	*T. longibrachiatum*	Hematologic malignancy	Pulmonary infection	VOR and CASP	Infection resolution
Kviliute et al. [41]	2008	Lithuania	*T. citrinoviride*	Acute myeloid leukemia	Pulmonary infection	AmB	Infection resolution
Lagrange-Xélot et al. [42]	2008	France	*Trichoderma longibrachiatum*	AIDS	Pulmonary infection	AmB and VOR	Infection resolution
Trabelsi et al. [43]	2010	Tunisia	*Trichoderma* spp.	Renal transplantation	Skin abscess and intertrigo	VOR	Infection resolution
Santillan Salas et al. [44]	2011	USA	*T. longibrachiatum*	Cardiac pediatric surgery	Mediastinitis and peritonitis	CASP and subsequently VOR and intraperitoneal AmB	Infection resolution
Rodríguez Peralta et al. [45]	2013	Spain	*T. longibrachiatum*	Home parenteral nutrition patient	Endocarditis	CASP	Infection resolution
Molnár-Gábor et al. [46]	2013	Hungary	*T. longibrachiatum*	Immunocompetent patient	Sinusitis	Topical AmB + surgical debridement	Infection resolution
Festuccia et al. [47]	2014	Italy	*Trichoderma* spp.	Auto-HSCT	Pulmonary infection	VOR	Infection resolution
Tascini et al. [48]	2016	Italy	*T. longibrachiatum*	ICD implantation complicated by PNX	Endocarditis	VOR and subsequently AmB	Infection resolution
Akagi et al. [49]	2017	Japan	*T. longibrachiatum*	Aplastic anemia	Pulmonary infection	AmB	Infection resolution
Carlson et al. [50]	2018	USA	*Trichoderma* spp.	Capd	Peritonitis	AmB and VOR	Infection resolution
Román-Soto et al. [51]	2019	Spain	*T. longibrachiatum*	Allo-HSCT	Necrotic ulcers	VOR, CASP + ulcer debridment	Infection resolution
Sautour et al. [52]	2019	France	*T. longibrachiatum*	Acute myeloid leukemia	Pulmonary infection	VOR + CASP	Infection resolution
Recio et al. [53]	2019	Spain	*T. longibrachiatum*	Pericardiectomy after lung transplant	Pericarditis	ANID and subsequently isavuconazole	Death
Bachu et al. [54]	2020	Turkey	*Trichoderma* spp.	Capd	Peritonitis	ANID and subsequently AmB	Death
Zhou [55]	2020	China	*T. longibrachiatum*	Lung cancer	Pulmonary infection	VOR	Infection resolution
Georgakopoulou et al. [56]	2021	Greece	*T. longibrachiatum*	Diabetes mellitus	Pulmonary infection	AmB	Infection resolution
Al-Shehri et al. [57]	2021	Saudi Arabia	*Trichoderma* spp.	Acute myeloid leukemia	Endophthalmitis	AmB + surgery	Infection resolution

AIDS: Acquired immunodeficiency syndrome, Allo-HSCT: Allogeneic Hematopoietic Stem Cell Transplant, AmB: Amphothericine B, ANID: Anidulafungin, CASP: Caspofungin, CAPD: continuous ambulatory peritoneal dialysis, FLU: fluconazole, 5-FC: 5-Flucytosine, KET: Ketoconazole, ITR: itraconazole, VOR: Voriconazole.

**Table 2 jof-09-00195-t002:** Summary of diagnostic approach and antifungal susceptibility testing.

Author	Clinical Manifestation	Specimen	Immunosuppression	Chemotherapy	Time from SOT/ CT	AST	Amb	Vor MIC	Casp	ITR	FLC	KET
							MIC	(mcg/mL)	MIC	MIC	MIC	MIC
							(mcg/mL)		(mcg/mL)	(mcg/mL)	(mcg/mL)	(mcg/mL)
Robertson et al. [16]	Disseminated	NA	No	No	NA	No						
Escudero Gil et al. [17]	TRP	BAL	NA	NA	NA	No						
Escudero Gil et al. [17]	TRP	BAL	NA	NA	NA	No						
Loeppky et al. [18]	Peritonitis	Peritoneal fluid	No	No		Yes	0.78					1.56
Ragnaud et al. [19]	Peritonitis	Peritoneal fluid	No	No		Yes	R			S		S
Jacobs et al. [20]	Peritonitis	Surgical debridement	Yes	No	21 days	Yes	3.1			1.6	25	0.8
Imokawa et al. [21]	TRP	BAL	No	No		No						
Tanis et al. [22]	Peritonitis	Peritoneal fluid	No	No	6 years	No						
Gautheret et al. [23]	Disseminated	BAL, skin biopsy	Yes	No	1 year	Yes	0.09			0.18	25	
Seguin et al. [24]	Brain abscess	Brain specimen	Yes	No	102 days	Yes	2.5			1.25	12.5	S
Kawaguchi et al. [25]	TRP	BAL	Yes	No	NA	No						
Guiserix et al. [26]	Peritonitis	Peritoneal fluid	No	No		No						
Campos-Herrero et al. [27]	Peritonitis	Peritoneal fluid	No	No		Yes	4.0			1.28		1.0
Munoz et al. [28]	Skin infection	Biopsy	Yes	No	8 mo	Yes	2.0			2.0	≥64	
Bren et al. [29]	Peritonitis	Peritoneal fluid	No	No		No						
Furukawa et al. [30]	Sinusitis	Biopsy	Yes	No	6 mo	Yes	0.58			0.035	40	
Richter et al. [31]	Disseminated	Biopsy	Yes	No	28 days	Yes	2.0			1.0		
Guarro et al. [32]	Brain abscess	Biopsy	Yes	No	1 mo	Yes	2.0			32	128	8
Rota et al. [33]	Peritonitis	Peritoneal fluid	No	No		No						
Bustamante-Labarta et al. [34]	Endocarditis	Biopsy	No	No		No						
Amato et al. [35]	Encephalitis	CSF	Yes	No	NA	No						
Chouaki et al. [36]	Perihepatic hematoma	Abscess aspirate	Yes	No	9 mo	Yes	>1	0.25		16		
Chouaki et al. [36]	TRP	BAL	Yes	No	few days	No						
Myoken et al. [37]	Stomatitis	Biopsy	Yes	No	27 days	Yes	0.5	0.5		≥32	≥64	
Esel et al. [38]	Peritonitis	Peritoneal fluid	No	No		Yes	0.5			1.5	2	1.5
De Miguel et al. [39]	TRP	BAL	Yes	No	12 days	Yes	0.25	2		8		
Alanio et al. [40]	TRP	BAL	Yes	No	4 days	Yes	0.5	1	0.5			
Kviliute et al. [41]	TRP	BAL	Yes	No	27 days	Yes	>32	4	0.125	>32		
Lagrange-Xélot et al. [42]	TRP	BAL	Yes	No	2 mo	Yes	1	0.5	0.5	2	>64	
Trabelsi et al. [43]	Skin infection	Byopsy	NA	No		Yes	1	0.5	0.5			
Santillan Salas et al. [44]	Peritonitis	Biopsy	No	No		Yes		≥1	≤0.5			
Rodríguez Peralta et al. [45]	Endocarditis	Peritoneal fluid	No	No		No						
Molnár-Gábor et al. [46]	Sinusitis	Biopsy	No	No		Yes	0,5	0.5	0.25			
Festuccia et al. [47]	TRP	BAL	Yes	No	10 days	Yes		0.125	0.047	>256	>32	
Tascini et al. [48]	Endocarditis	Catheter tip	No	No		Yes	2	0.5	1			
Akagi et al. [49]	TRP	BAL	Yes	No	151 days	No						
Carlson et al. [50]	Peritonitis	Peritoneal fluid	No	No		No						
Román-Soto et al. [51]	Necrotic ulcers	Skin biopsy	Yes	No	NA	Yes	1.5	0.23	0.047	0.23		
Sautour et al. [52]	TRP	BAL	Yes	No	11 days	Yes	2	0.19	0.064	>32		
Recio et al. [53]	Pericarditis	BAL	Yes	No	2 mo	Yes	1	2	1			
Bachu et al. [54]	Peritonitis	NA	No	No		No						
Zhou [55]	TRP	Biopsy	Yes	No	0 days	Yes	4	1	≤0.03	>16	>64	
Georgakopoulou et al. [56]	TRP	Pleural fluid	No	No		Yes	S	R	R	R	R	
Al-Shehri et al. [57]	Endophthalmitis	Vitreous tap	Yes	No	2 weeks	Yes	0.5	16	0.5			

AmB: Amphothericine B, AST: antifungal susceptibility testing, CASP: Caspofungin, FLU: fluconazole, CT: chemotherapy, KET: ketoconazole, MIC: minimal inhibitory concentration, ITR: itraconazole, VOR: Voriconazole, Gal: galactomannan.

## Data Availability

Not applicable.

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
