# Peer review of "Trichoderma spp.-Related Pneumonia: A Case Report in Heart–Lung Transplantation Recipient and a Systematic Literature Review"

_jof, 2023, doi:10.3390/jof9020195_

Round 1

Reviewer 1 Report

In this paper, Burzio, et al, describe a clinical case of a patient with presumed Trichoderma pneumonia, followed by a systematic literature review of Trichoderma human infections. This review is pertinent, because it includes up to 44 papers (though namely case reports), which is one of the largest collections of human cases of infection available. However, the paper's impact could be improved to increase clarity for the reader, including some of the recommendations listed below.

1. The clinical background discusses some patient information from prior to transplant that is likely not relevant to the paper and can be cut (page 2, "...first admitted due to bilateral pneumothoraces treated with chest drains.")

2. The authors comment on page 2 about their local antimicrobial prophylaxis, including inhaled amphotericin. However, more detail on that protocol would be helpful here (what other antimicrobials, any other antifungals, how frequently is ampho B dosing, etc).

3. There are some mild areas of confusion in the paper for the reader that may be due to some English translation concerns, one example is page 2, "Even if the patient was successfully weaned...". This sentence is confusing and should be edited.

4. Were the authors able to pursue any further fungal diagnostics into the Trichoderma spp identified?

5. It would be helpful for the authors to clarify the reason for using the combination therapy with voriconazole/caspofungin, given that the literature suggests Ampho B was the most frequently used regimen. The authors get into this some in the Discussion on Page 9, but could perhaps provide more detail. Was the inhaled amphotericin continued when the systemic voriconazole and echinocandin were started? Where they worried about resistance because the infection developed while the patient was on inhaled amphotericin?

6. The end of the section on the clinical case (pages 2 and 3) are slightly contradictory which can be confusing to the reader. The authors write that there was resolution of pneumonia by clinical and radiographic parameters and negative BAL, however in the next sentence they state the patient remained still critically ill and ventilated. It would be helpful to clarify this and identify what the overall outcome of the patient was. Did all antifungals stop at 34 days? That would be on the shorter end for fungal pneumonia courses, particularly in a transplant case, which are often closer to 6-12 weeks.

7. Do the authors have any epidemiologic data on the donor or recipient that might suggest what led them to develop such a significant pulmonary Trichoderma infection? This might be a helpful piece of information for the reader so they know when to have a higher suspicion for this particular pathogen.

8. The language on page 4 describing the review paper characteristics and geographic prevalence is confusing to read and may need some editing for clarification of the demographics the authors are describing.

9. The first paragraph on page 5 of the dates specific to each country of infection is somewhat redundant with the Table 1 just below -- that paragraph can likely be cut as I am not sure that it adds much to the paper that the extensive Table doesn't already cover.

10. In Table 1, it might also be helpful to add a column listing the bodily fluid that revealed the positive culture (ie, peritoneal fluid, BAL fluid, blood culture, wound culture, etc). This is suggested by the "Clinical manifestation" column, but not confirmed. Given the limitations in fungal diagnostics for more rare molds, it would be helpful to understand which sites had a higher culture yield.

11. Page 8 describes that patients who developed Trichoderma were 'immunocompromised' (16 cases) but then also lists other categories of patients that are also immunocompromised (hematologic, transplant, AIDS). Can the authors clarify what is different about the 'immunocompromised' cases from the others? This may also help clarify a sentence in the second paragraph of the Discussion on Page 8, that states that this infection occurs 'only' in immunocompromised patients, though the abstract describes that one case was not immunocompromised.

12. Do the cases reviewed ever go into antifungal testing of Trichoderma? The authors list some references that included AST data [36,39,55,68] but don't get into details about what that AST data demonstrated aside from one sentence listing reference 14. I understand that AST is not routinely done for this pathogen, but may have been done sporadically in other cases and that data could offer some insight for future treatment recommendations.

Author Response

In this paper, Burzio, et al, describe a clinical case of a patient with presumed Trichoderma pneumonia, followed by a systematic literature review of Trichoderma human infections. This review is pertinent, because it includes up to 44 papers (though namely case reports), which is one of the largest collections of human cases of infection available. However, the paper's impact could be improved to increase clarity for the reader, including some of the recommendations listed below.

  1. The clinical background discusses some patient information from prior to transplant that is likely not relevant to the paper and can be cut (page 2, "...first admitted due to bilateral pneumothoraces treated with chest drains.")

****Response: We appreciate the Reviewer’s insightful comment. We agree with the Reviewer’s point, we deleted this paragraph as it is not relevant from a reader's perspective.

  1. The authors comment on page 2 about their local antimicrobial prophylaxis, including inhaled amphotericin. However, more detail on that protocol would be helpful here (what other antimicrobials, any other antifungals, how frequently is ampho B dosing, etc).

****Response:  We appreciate the Reviewer’s question and the opportunity to clarify.  We incorporated the description of our hospital’s protocol as it follows:

“Antimicrobial prophylaxis was started according to our local protocol, including inhaled amphotericin B (Amphotericin B lipid complex, 20mg once daily) and oral nystatin (1 MIE oral drops, TID). Our protocol also included antibacterial prophylaxis (ceftazidime, IV 1gr TID, for the first 72hs), Pneumocystis jirovecii prophylaxis (Trimetoprim-Sulfamethoxazole, PO, once daily) and citomegalovirus (CMV) prophylaxis ac-cording to CMV Status (Valganciclovir, PO, 900mg once daily).”

  1. There are some mild areas of confusion in the paper for the reader that may be due to some English translation concerns, one example is page 2, "Even if the patient was successfully weaned...". This sentence is confusing and should be edited.

****Response:  We apologize for the lack of clarity. We conducted a careful review of the grammar and corrected the sentence as indicated by the Reviewer.

  1. Were the authors able to pursue any further fungal diagnostics into the Trichoderma spp identified?

****Response:  We thank the Reviewer for asking this question.  We actually conducted some diagnostics by consecutive evaluation of fungal infection. Initially, our team performed repeated bronchoscopies to obtain a sample for AST. Unfortunately, the sample was not adequate. We also conducted a search for galactomannan on BAL. As the Reviewer rightly reported, we then expanded the description of the diagnostic investigation as follows:

“Repeated BALs were performed, as well as culture examination of mucous plug; mi-croscopic examination confirmed Trichoderma spp. and hyphal growth in all the samples, but out laboratory was unable to grow viable cultures, identify Trichoderma species or perform antimicrobial susceptibilities testing (AST). […]Parenteral antifugal therapy was therefore discontinued, and prophylaxis with inhaled Ampho-tericin B alone was maintained. Following surveillance cultures, including subsequent BALs and BAL-Galactomannan assays, were negative for Trichoderma or fungal detection. “

  1. It would be helpful for the authors to clarify the reason for using the combination therapy with voriconazole/caspofungin, given that the literature suggests Ampho B was the most frequently used regimen. The authors get into this some in the Discussion on Page 9, but could perhaps provide more detail. Was the inhaled amphotericin continued when the systemic voriconazole and echinocandin were started? Where they worried about resistance because the infection developed while the patient was on inhaled amphotericin?

****Response:  We apologize for leaving out these important details from the discussion. We added in the text our choice’s explanation as it follows:

“Nevertheless, AST for Trichoderma infections is being widely utilized in clinical practice (Table 2), often assuming clinical breakpoints for unspecified mold infection. By our literature review suggest that Trichoderma maintain high in-vitro resistance for azoles, showing MIC often >32μg/mL. High (>1μg/mL) MIC values are often reported for voriconazole and amphotericin B as well. Echinocandins exhibited instead MIC<1μg/mL in most of the reported cases. These results agree with a previous study regarding in vitro susceptibilities of Trichoderma isolated, including both human and animal samples (however, 40% of samples were collected from human respiratory tract), which reported mostly low MIC for voriconazole and echinocandins [14].”

  1. The end of the section on the clinical case (pages 2 and 3) are slightly contradictory which can be confusing to the reader. The authors write that there was resolution of pneumonia by clinical and radiographic parameters and negative BAL, however in the next sentence they state the patient remained still critically ill and ventilated. It would be helpful to clarify this and identify what the overall outcome of the patient was. Did all antifungals stop at 34 days? That would be on the shorter end for fungal pneumonia courses, particularly in a transplant case, which are often closer to 6-12 weeks.

*****RESPONSE:  We thank the Reviewer for helping us clarify this part of the manuscript. We agree that the case was not carefully described because of the great clinical complexity. The patient over time acquired significant respiratory muscle weakness, which did not allow a liberation from mechanical ventilation. For this reason, the course was further complicated months after the resolution of the patient's clinical situation. We added the following sentences to the end of the description of the case report:

“The patient suffered ICU-Acquired Weakness and difficult weaning from mechanical ventilation. Her clinical course was further complicated by MDR Klebsiella pneumoniae colonization and surgical site infection. Relapsing septic shock despite surgical debridement and end-stage renal failure ensued. The patient died from Klebsiella related septic shock 6 months after transplantation.

While neither Trichoderma nor respiratory failure were direct causes of death, TRP was an early complication that required prolonged ICU length of stay, thus hindering the postoperative clinical course.”

  1. Do the authors have any epidemiologic data on the donor or recipient that might suggest what led them to develop such a significant pulmonary Trichoderma infection? This might be a helpful piece of information for the reader so they know when to have a higher suspicion for this particular pathogen.

****Response:  We thank the Reviewer for the excellent suggestion.  As interesting as the point made by the Reviewer is, Italian Law guarantees full anonimity for organ donors (as, for examples, keeping donor and recipient records strictly separated), so we cannot disclose any detailled information about the donor, and neither discuss about it. We added the following sentence:

“no specific environmental factor could be found in medical history of both the organ donor and the patient other than prolonged pre-transplantation hospitalization of the recipient.”

  1. The language on page 4 describing the review paper characteristics and geographic prevalence is confusing to read and may need some editing for clarification of the demographics the authors are describing.

****Response:  We thank the Reviewer for the suggestion which we have carefully considered. We decided to delete the paragraph by integrating it partially with the previous paragraph (3.1. Search results) and removing information that the Reviewer also defined as redundant since they were previously described in the Table 1.

  1. The first paragraph on page 5 of the dates specific to each country of infection is somewhat redundant with the Table 1 just below -- that paragraph can likely be cut as I am not sure that it adds much to the paper that the extensive Table doesn't already cover.

*****RESPONSE:  We are grateful to the Reviewer for the comment that allowed us to clarify the point. As we mentioned before, we deleted the paragraph, making sure that no doubled information was reported in the main text.

  1. In Table 1, it might also be helpful to add a column listing the bodily fluid that revealed the positive culture (ie, peritoneal fluid, BAL fluid, blood culture, wound culture, etc). This is suggested by the "Clinical manifestation" column, but not confirmed. Given the limitations in fungal diagnostics for more rare molds, it would be helpful to understand which sites had a higher culture yield.

****Response:  We thank the Reviewer for the excellent suggestion.  As the Reviewer rightly pointed out, we so appreciated the advice that we have created a new table (Table 2) with additional relevant information, to better understand Trichoderma infections. During the review, some data were also corrected, as highlighted.

  1. Page 8 describes that patients who developed Trichoderma were 'immunocompromised' (16 cases) but then also lists other categories of patients that are also immunocompromised (hematologic, transplant, AIDS). Can the authors clarify what is different about the 'immunocompromised' cases from the others? This may also help clarify a sentence in the second paragraph of the Discussion on Page 8, that states that this infection occurs 'only' in immunocompromised patients, though the abstract describes that one case was not immunocompromised.

****Response:  We thank the Reviewer for the suggestion which we have carefully considered.  As suggested, we added a paragraph devoted to risk factors related to a Trichoderma infection. To make this paragraph we used the new literature review that we conducted when creating Table 2.

“Specific risk factors for Trichoderma infection are difficult to detect due to paucity of cases. The literature review confirmed that Trichoderma spp. infections occur mostly in immunocompromised patients (Table 1). Our patient showed several risk factors for invasive aspergillosis, as her medication regimen included steroid for immunosuppressive therapy, and Anti-Thymocyte Immunoglobulin as induction therapy; notably, no specific environmental factor could be found in medical history of both the organ donor and the patient other than prolonged pre-transplantation hospitalization of the recipient.”

  1. Do the cases reviewed ever go into antifungal testing of Trichoderma? The authors list some references that included AST data [36,39,55,68] but don't get into details about what that AST data demonstrated aside from one sentence listing reference 14. I understand that AST is not routinely done for this pathogen, but may have been done sporadically in other cases and that data could offer some insight for future treatment recommendations.

****Response:  We thank the Reviewer for giving us the opportunity to clarify this measurement.  To facilitate the evaluation of AST in the considered cases of Trichoderma infection, we created a special section in Table 2, added a new paragraph in the results (3.3 Diagnostic Tests and Antifungal susceptibility testing), and discussed this new revision as follows:

“Nevertheless, AST for Trichoderma infections is being widely utilized in clinical practice (Table 2), often assuming clinical breakpoints for unspecified mold infection. By our literature review suggest that Trichoderma maintain high in-vitro resistance for az-oles, showing MIC often >32μg/mL. High (>1μg/mL) MIC values are often reported for voriconazole and amphotericin B as well. Echinocandins exhibited instead MIC<1μg/mL in most of the reported cases. These results agree with a previous study regarding in vitro susceptibilities of Trichoderma isolated, including both human and animal samples (however, 40% of samples were collected from human respiratory tract), which reported mostly low MIC for voriconazole and echinocandins [14].”

Reviewer 2 Report

The study authors present a case of a recent heart-lung transplant recipient who was subsequently diagnosed with Trichoderma pneumonia. They perform a narrative systematic review to evaluate prior reports of human infections.

Overall, I believe the case report is well written with Table 1 providing an exhaustive summary of the known infections in humans. I just have a few additions to better improve certain aspects of the manuscript.

Major Comments

1. The study authors report positive cell culture for Trichoderma species as well as hyphal growth in the BAL cytology specimen. Was further workup for species-level identification performed? It would be helpful to provide that information.

2. It is interesting to note the fungal infection occurring during the period of initial intense immunosuppression while on inhaled amphotericin B (AMB) prophylaxis. Was this reported to occur in other solid organ transplant (SOT) recipient cases with sinopulmonary infections as well? Additionally, what was the timeframe of infections occurring other SOT recipients? The study authors could consider including an additional Table describing certain clinical points of interest in relation to SOT recipients – such as age of the patient, timing of infection from transplant, immunosuppressive regimen at time of infection.

3. The study authors note that antifungal susceptibility testing was not performed and that is technically difficult. However, it is interesting to note that some patients received systemic AMB for therapy. Without antifungal susceptibility testing, it is difficult to ascertain whether the infection in the current patient was a breakthrough infection on AMB – the authors should consider raising this point. Additionally, given that the post-transplant antimicrobial prophylaxis regimens can vary significantly between centers and countries, it would be important to raise awareness amongst clinicians about the possibility of fungal infections occurring during prophylaxis and to tailor their prophylaxis regimens based on their local fungal epidemiology.

Minor Comments

1. Page 2 of 14, line 48-49 – the line “with the development of resistance, deserving now antifungal stewardship strategies [8,9]” appears to be slightly disjointed. Did the study authors mean “new antifungal stewardship strategies”? Please rectify the line accordingly.

2. Page 2 of 14, line 83-84 – the authors report “Even if the patient was successfully weaned from the inotropic support end and extubated on postoperative day 5, the clinical conditions worsen on the ninth postoperative day due to the onset of moderate hypoxemia and clinical signs of pump failure”. I believe they might have meant to use “after” instead of “if” and “worsened” instead of “worsen”. Please rectify accordingly.

3. At multiple points in the text (for example, page 1 of 14, line 42-43), the genus names of fungal species should be italicized.

4. The study authors raise multiple important points in the Conclusions, but they may be better in the Discussion. I would suggest to try to trim the Conclusions to just a few key points.

Author Response

We are grateful for the opportunity to revise and resubmit our manuscript, and we appreciate the time that you and the reviewers spent critically evaluating our manuscript to help us improve the delivery of our message. We have addressed each comment point by point in the rebuttal section and have revised the manuscript by incorporating Editor and reviewer feedback.  The revised manuscript contains 5333 words with one new table and has been read and approved by all co-authors. We also seized the opportunity to make some minor corrections and to clarify some points, according to reviewers suggestions; all changes are highlighted in the manuscript. Thank you again for the consideration of our manuscript for possible publication.

The study authors present a case of a recent heart-lung transplant recipient who was subsequently diagnosed with Trichoderma pneumonia. They perform a narrative systematic review to evaluate prior reports of human infections.

Overall, I believe the case report is well written with Table 1 providing an exhaustive summary of the known infections in humans. I just have a few additions to better improve certain aspects of the manuscript.

****Response:  We thank the Reviewer for the positive comments!

Major Comments

  1. The study authors report positive cell culture for Trichoderma species as well as hyphal growth in the BAL cytology specimen. Was further workup for species-level identification performed? It would be helpful to provide that information.

****Response:  We appreciate the Reviewer’s comments.  We agree with the Reviewer regarding the point he correctly pointed out as incomplete.  We conducted some diagnostics performing repeated bronchoscopies to obtain a sample for AST. Unfortunately, the sample was not adequate. We conducted a search for galactomannan on BAL. Considering the lack of these information we have integrated into the text as follows:

“Repeated BALs were performed, as well as culture examination of mucous plug; microscopic examination confirmed Trichoderma spp. and hyphal growth in all the samples, but out laboratory was unable to grow viable cultures, identify Trichoderma species or perform antimicrobial susceptibilities testing (AST). […] Parenteral antifugal therapy was therefore discontinued, and prophylaxis with inhaled Amphotericin B alone was maintained. Following surveillance cultures, including subsequent BALs and BAL-Galactomannan assays, were negative for Trichoderma or fungal detection.”

  1. It is interesting to note the fungal infection occurring during the period of initial intense immunosuppression while on inhaled amphotericin B (AMB) prophylaxis. Was this reported to occur in other solid organ transplant (SOT) recipient cases with sinopulmonary infections as well? Additionally, what was the timeframe of infections occurring other SOT recipients? The study authors could consider including an additional Table describing certain clinical points of interest in relation to SOT recipients – such as age of the patient, timing of infection from transplant, immunosuppressive regimen at time of infection.

****Response:  We thank the Reviewer for the suggested edit, which we have incorporated into a new table (Table 2). We are very grateful for the opportunity to integrate this new information into our work. We decided to incorporate the underlined information from the Reviewer in his major comments number 2 and number 3. After a further review of each case report, we added the time of infection from transplantation or any current chemotherapy, immunosuppressive therapy, or chemotherapy, the type of specimen on which the microbiological analysis was performed, and reported - when indicated - AST for the major antifungals mentioned in our review. During the review, some data were also corrected, as highlighted.

  1. The study authors note that antifungal susceptibility testing was not performed and that is technically difficult. However, it is interesting to note that some patients received systemic AMB for therapy. Without antifungal susceptibility testing, it is difficult to ascertain whether the infection in the current patient was a breakthrough infection on AMB – the authors should consider raising this point. Additionally, given that the post-transplant antimicrobial prophylaxis regimens can vary significantly between centers and countries, it would be important to raise awareness amongst clinicians about the possibility of fungal infections occurring during prophylaxis and to tailor their prophylaxis regimens based on their local fungal epidemiology.

****Response:  We appreciate the Reviewer’s excellent suggestions. We covered this point by responding to the previous comment.

Minor Comments

  1. Page 2 of 14, line 48-49 – the line “with the development of resistance, deserving now antifungal stewardship strategies [8,9]” appears to be slightly disjointed. Did the study authors mean “new antifungal stewardship strategies”? Please rectify the line accordingly.

****Response:  We agree with the Reviewer’s suggestion and have made this change.

  1. Page 2 of 14, line 83-84 – the authors report “Even if the patient was successfully weaned from the inotropic support end and extubated on postoperative day 5, the clinical conditions worsen on the ninth postoperative day due to the onset of moderate hypoxemia and clinical signs of pump failure”. I believe they might have meant to use “after” instead of “if” and “worsened” instead of “worsen”. Please rectify accordingly.

****Response:  We thank the Reviewer for the suggested edit which we have incorporated.

  1. At multiple points in the text (for example, page 1 of 14, line 42-43), the genus names of fungal species should be italicized.

****Response:  We thank the Reviewer for the suggested edits which we have incorporated.

  1. The study authors raise multiple important points in the Conclusions, but they may be better in the Discussion. I would suggest to try to trim the Conclusions to just a few key points.

****Response:  We thank the Reviewer for the supportive comment and for pointing this out.  We agree 100% that the conclusion must be just few key points. Thus, we cut the conclusion as it follows:

“Although rare, IFI caused by uncommon fungi, such as Trichoderma spp. and other NAIMI, are increasingly reported in critically ill patients, especially in SOT recipients, presenting challenges in terms of clinical suspicion, diagnostic methods, and treatment options.

Large and definitive clinical trials involving Trichoderma human infection are unlikely due to the paucity of cases. The topic of re-emerging fungal infections also makes the possibility of an individual-patient meta-analysis difficult because of the great differences in the geographic and historical context.

 In case of future reports in the literature, we suggest improving the available evidence by reporting in more detail the MIC for antifungals, days of therapy, the outcome, and the concomitant complications developed by the patient. “

Round 2

Reviewer 1 Report

I thank the authors for addressing the Reviewer comments appropriately.